# Integrated Multi-Omics Analysis Reveals Immune and Metabolic Dysregulation in a Restraint Stress-Induced Depression Model

**DOI:** 10.3390/biomedicines13092183

**Published:** 2025-09-06

**Authors:** Ziying Wang, Xiangyu Wang, Yuting Li, Qian Zhao, Zhaohui Lan, Weidong Li

**Affiliations:** 1Key Laboratory for the Genetics of Development and Neuropsychiatric Disorders, Institute of Psychology and Behavioral Sciences, Bio-X Institutes, Shanghai Jiao Tong University, Shanghai 200240, China; wangziying@sjtu.edu.cn (Z.W.); wangxiangyu12345@alumni.sjtu.edu.cn (X.W.); liyuting1@sjtu.edu.cn (Y.L.); zhaoqian323@sjtu.edu.cn (Q.Z.); 2Brain Health and Brain Technology Center, Global Institute of Future Technology, Shanghai Jiao Tong University, Shanghai 200240, China

**Keywords:** depression, restraint, transcriptomics, proteomics, clemastine

## Abstract

**Background:** Major depressive disorder (MDD) is a prevalent and disabling psychiatric illness with complex etiologies involving both genetic and environmental factors. While environmental stress is a known risk factor of MDD, the molecular mechanisms linking stress exposure to persistent depressive phenotypes remain incompletely understood. **Methods**: We established a 24-hour restraint stress-induced depression model in mice and performed integrated transcriptomic and proteomic analyses of the medial prefrontal cortex (mPFC) to investigate stress-related molecular alterations. **Results**: Behavioral assessments confirmed persistent depression-like phenotypes, including anhedonia and behavioral despair, lasting up to 35 days post-stress. RNA sequencing identified differentially expressed genes related to dopaminergic signaling and oxidative stress. Proteomic analysis identified 105 differentially expressed proteins involved in immune response and energy metabolism. Integrated multi-omics analysis highlighted convergent disruptions in immune regulation, metabolism, and epigenetic processes. Notably, clemastine exerts its antidepressant-like effects in part by mitigating neuroinflammation and preserving mitochondrial function. **Conclusions**: These findings provide novel insights into the molecular basis of stress-induced depression and suggest that clemastine is a potential therapeutic candidate.

## 1. Introduction

Major depressive disorder (MDD) is one of the most prevalent and disabling psychiatric disorders, affecting approximately 5% of adults globally each year [1,2]. Its etiology and pathogenesis are multifactorial, involving both genetic factors and environmental stimulations [3]. Despite increasing research efforts, the pathophysiological mechanisms underlying MDD remain incompletely understood. Evidence from clinical studies in MDD patients and preclinical animal models has revealed that MDD is associated with alterations in brain structure and functional connectivity, dysregulation in neurotransmitter systems, and an imbalance between excitatory and inhibitory (E/I) signaling [4,5,6]. In addition, growing evidence shows neuroimmune dysregulation in depression [7,8]. The cytokine hypothesis proposes that stressors promote depressive behavior by elevating pro-inflammatory cytokines and activating cell-mediated immune responses [9].

The medial prefrontal cortex (mPFC), a key region for emotional regulation and stress response, is highly affected in animal models of chronic stress. Chronic stress disrupts glutamatergic projections within the mPFC and alters the expression of NMDA receptor subunits, thereby modulating neuronal excitability [10]. In our previous study, we established a 24-hour restraint stress-induced depression model, where mice exhibited reduced whole-brain glucose uptake, impaired neurogenesis, and downregulated expression of myelination-related genes in the prefrontal cortex and hippocampus 35 days after the restraint [11,12]. Although traditional depression models often require prolonged stress exposure, our findings demonstrated that an acute restraint stress paradigm is sufficient to induce persistent depression-like phenotypes in mice. However, the molecular mechanisms that bridge acute stress exposure and long-lasting depressive phenotypes, particularly in the mPFC, remain poorly understood.

Recent advances in integrative multi-omics strategies offer comprehensive insights into disease-associated molecular networks and facilitate the identification of potential biomarkers and therapeutic targets [13,14]. In this study, we performed transcriptomic and proteomic profiling of mPFC tissue from 24-hour restraint stress-induced depressive mice. We identified dysregulated pathways involved in dopaminergic signaling, oxidative stress, immune function, and mitochondrial metabolism, consistent with our earlier findings [11]. Notably, clemastine treatment reversed stress-induced changes in inflammation, epigenetic regulation, and metabolic homeostasis, underscoring its therapeutic potential. These findings provide new mechanistic insights into acute stress-induced depression and highlight clemastine as a promising intervention.

## 2. Materials and Methods

### 2.1. Animals

Adult male C57/BL6 mice (8–12 weeks old) were purchased from Beijing Vital River Laboratory Animal Technology Co., Ltd. (Beijing, China). All mice were housed in temperature-fixed (22 ± 2 °C), humidity-controlled chambers, and sufficient food and water were administered daily. All animal experiments were performed according to the guidelines approved by the University Committee on Animal Care and Use of Shanghai Jiao Tong University, China. The Animal Protocol number is A2023190-001.

All experiments were performed on adult male C57BL/6 mice. Males were selected to minimize variability associated with the estrous cycle in females, which can affect stress responses, neuroimmune activity, and behavior. This approach allowed us to reduce within-group variance for this initial mechanistic study.

### 2.2. Study Design

Experiment 1: Chronic Restraint Stress Model: Male mice were randomly assigned to two groups: Control group (*n* = 17): Standard housing without intervention. Restraint group (*n* = 16): Subjected to 24-hour acute restraint stress, followed by return to home cages. After 35 days, behavioral tests were conducted, including the sucrose preference test (SPT) and forced swim test (FST) to assess depressive-like phenotypes. Subsequently, the medial prefrontal cortex (mPFC) was dissected from 3 mice per group randomly for transcriptomic (RNA-seq) and proteomic profiling (LC-MS/MS). Paired samples from the same mice (*n* = 3/group) to minimise individual difference.

Experiment 2: Clemastine Intervention Study: To evaluate therapeutic effects, a separate cohort of male mice was divided into four groups: Control + Water (*n* = 10): Non-stressed, vehicle-treated; Control + Clemastine (*n* = 10): Non-stressed, clemastine-treated; Restraint + Water (*n* = 12): Stressed, vehicle-treated; Restraint + Clemastine (*n* = 12): Stressed, clemastine-treated. Clemastine or vehicle administration began 2 days post-restraint and continued for 35 days, followed by depressive-like behavior assessments. The mPFC of 3 mice per group was randomly chosen for proteomic analysis (LC-MS/MS).

Inclusion criteria for all experiments required mice to be healthy (no visible wounds or weight loss > 15%) and complete the full protocol (e.g., no early death during restraint). For omics analysis, we selected three biological replicates per group, consistent with common practice in exploratory omics studies [13,15]. To minimize inter-individual variability and increase robustness, paired samples from the same mice were used, and only high-quality RNA (RIN > 8) and protein samples were included based on pilot testing. These thresholds were defined before data collection based on pilot studies. Behavioral data were excluded if mice failed to explore both bottles in SPT or if tracking errors occurred in FST videos (manually verified).

Behavioral scoring was performed by observers blinded to group allocation. For RNA-seq and proteomic experiments, samples were processed and labeled under coded IDs. Library preparation, LC-MS/MS acquisition, and initial bioinformatics/statistical analyses were conducted without knowledge of group identity. Group codes were only revealed after completion of primary analyses.

### 2.3. Procedure for the 24-Hour Restraint Model

According to the previous study [11], the mice were restrained for 24 h in darkness without water and food, with only their heads able to move. The restraint devices have small ventilation and heat dissipation holes to ensure that the mice can breathe freely. Once the restraint ended, mice were put back into their home cages immediately, with access to food and water freely.

### 2.4. Drug Administration

Clemastine was administered in drinking water at 0.05 mg/mL starting 2 days after restraint and continued for 32 days. Solutions were freshly prepared daily, protected from light, and replaced every 24 h to ensure stability. Average water intake was ~4.5 mL/mouse/day, corresponding to ~10 mg/kg/day for a 22 g mouse. This dosing regimen was chosen based on previous studies demonstrating efficacy and tolerability of clemastine in rodent CNS models [16,17]. Body weight and home-cage activity were monitored throughout the study, and no sustained anticholinergic adverse effects were observed.

### 2.5. Forced Swim Test (FST)

Behavioral scoring (SPT/FST) used blinded observers.

FST was performed as previously described [12], tested mice were individually placed in a 2-L beaker containing clean water (25 ± 1 °C). The water depth was set as 20 cm to prevent the mice from touching the bottom with the tail or hind legs. Mouse behaviors were videotaped from the side for 6 min. The immobility time of the last 4 min were analyzed by Noldus-maze. Immobility was defined as the duration that animals remained floating or stationary.

### 2.6. Sucrose Preference Test (SPT)

SPT was carried out at the home cage of tested mice housed individually. To allow tested mice to adapt, two bottles of water were placed on the wire lid for 48 h and were replaced by two bottles of 1% sucrose for another 48 h before the test. Then, after 24 h water deprivation, two bottles containing water and 1% sucrose, respectively were placed for 24 h with a position switch every 12 h. The consumed water and 1% sucrose in the 24 h for each tested mouse were measured, and sucrose preference was calculated by (Consumed 1% sucrose)/(Consumed water + Consumed 1% sucrose) × 100%.

### 2.7. Transcriptomics Analysis

RNA sequencing was performed as previously described [18]. Total RNA was extracted using TRIzol reagent following lysozyme pretreatment, and purified RNA was quantified with an Agilent 2100 Bioanalyzer (Agilent Technologies, Santa Clara, CA, USA) and NanoDrop spectrophotometer (Thermo Fisher Scientific, Wilmington, DE, USA). RNA-seq libraries were prepared and sequenced on the Illumina HiSeq X TEN platform. Clean reads were aligned to the reference genome using HISAT2, and gene expression levels were quantified as fragments per kilobase of transcript per million mapped reads (FPKM). Differentially expressed genes (DEGs) were identified using edgeR (fold change ≥ 2, adjusted *p* ≤ 0.05). GO and KEGG enrichment analyses were performed to explore functional classifications and pathway associations of DEGs.

Gene set enrichment analysis (GSEA) was conducted in R using the clusterProfiler package (v4.16.0). All detected genes were included, and the ranked gene list was generated by ordering log2 fold change (log2FC) values from highest to lowest. The reference gene sets were obtained from the MSigDB mouse Hallmark collection (H, 50 sets, release v7.5.1, Broad Institute). Enrichment analysis was performed with 1000 permutations using the gene set permutation mode, and significance was assessed by normalized enrichment score (NES), nominal *p* value, and false discovery rate (FDR q) adjusted by the Benjamini–Hochberg procedure. Pathways with FDR *q* < 0.25 were considered significant, following GSEA recommendations.

### 2.8. Proteomics Analysis

Total protein was extracted using 8 M urea lysis buffer with protease inhibitors, followed by sonication and trichloroacetic acid precipitation. Protein concentration was determined using the BCA assay. After trypsin digestion and C18 desalting, peptides were fractionated by high-pH reversed-phase LC using an Ultimate 3000 HPLC system (Thermo Fisher Scientific, Waltham, MA, USA) equipped with an XBridge C18 column (Waters Corporation, Milford, MA, USA) with a 40-min linear gradient, and subsequently analyzed on an EASY-nLC 1200 nanoLC system coupled to an Orbitrap Fusion Lumos mass spectrometer (both from Thermo Fisher Scientific, Waltham, MA, USA). Separation was performed on an Acclaim PepMap C18 column (75 μm × 25 cm) with a 120 min gradient (5–35% acetonitrile, 0.1% formic acid). DDA and DIA were acquired with standard Orbitrap settings (MS1 resolution 120,000; MS2 resolution 15,000–30,000; AGC targets 4 × 10^5^ to 1 × 10^6^ ions; stepped NCE 32 ± 5). Raw data were searched in Spectronaut X against UniProt mouse (release 2023_01) with a reversed decoy, controlling FDR ≤ 1% at the PSM, peptide, and protein levels. LFQ intensities were log2 transformed, missing values were imputed (downshifted normal distribution in Perseus), normalized by median centering, and batch effects were corrected using ComBat.

Functional annotation of identified proteins was performed using UniProt-GOA (release 2023_06)for Gene Ontology (GO) terms, InterProScan (version 5.61-93.0, EMBL-EBI) for domain analysis, and KEGG (accessed on March 2025) for pathway enrichment. Enrichment analysis of differentially expressed proteins was conducted using Fisher’s exact test, with *p* < 0.05 considered statistically significant.

### 2.9. Statistical Analysis

Numerical variables are presented as the mean ± SEM and were analyzed using GraphPad Prism software (version 8.3.1). Before statistical analysis, the distribution of the data was assessed for normality using the Shapiro—Wilk test. Unpaired two-tailed *t* tests were used to evaluate the significant differences between the two groups. The comparison of data among multiple groups was performed with one-way ANOVA, followed by the Bonferroni post hoc test. *p* values < 0.05 were considered statistically significant (* *p* < 0.05, ** *p* < 0.01, *** *p* < 0.001).

## 3. Results

### 3.1. Transcriptomic Profiling Reveals Dopaminergic, Oxidative, and Metabolic Pathway Disruptions Following Restraint Stress

Notably, mice exhibited reduced sucrose preference and increased immobility in the forced swim test (FST) 35 days post-stress (Figure 1a,b). To investigate the underlying molecular mechanisms, we conducted RNA sequencing of mPFC tissue from control and stressed mice. Differential expression analysis identified 59 genes (fold change ≥ 2.0, *p* < 0.05), including 25 up-regulated and 34 down-regulated DEGs (Figure 1c,d, Appendix A). Gene Ontology (GO) enrichment analysis showed that up-regulated genes were involved in dopaminergic signaling and synaptic plasticity, including dopamine D2 receptor (Drd2), adenosine A2A receptor (Adora2a), and regulator of G protein signaling 9 (Rgs9), suggesting impaired reward processing and cognitive regulation (Figure 1e,f). Down-regulated genes were enriched in oxidative stress response, hemoglobin complex, and oxygen transport pathways, including calprotectin (S100A8/A9), hemoglobin subunits (Hba1/Hbb), and heat shock protein 70 (Hsp70), indicating reduced antioxidant defense and possible tissue hypoxia (Appendix A).

To identify functionally enriched pathways associated with restraint stress, we performed gene set enrichment analysis (GSEA) on the genome-wide ranked gene expression data. The results revealed upregulation of pathways related to β-cell function (NES = 1.49, *p* < 0.05) and adipogenesis (NES = 1.31, *p* < 0.05), while hypoxia (NES = −1.89, *p* < 0.001), TNF-α/NF-κB signaling (NES = − 2.09, *p* < 0.001), and heme metabolism (NES = − 1.85, *p* < 0.001) pathways were significantly suppressed (Figure 1g,h, Appendix A). These findings suggest that acute restraint stress-induced depressive phenotypes may be accompanied by metabolic disturbances, suppressed inflammatory signaling, and impaired oxygen utilization, aligning with clinical observations of altered energy metabolism and neuroinflammatory features in patients with depression.

### 3.2. Proteomic Profiling Reveals Stress-Induced Dysregulation of Epigenetic Processes, Immune Responses, and Mitochondrial Metabolism

Proteomic analysis of the prefrontal cortex identified 7957 proteins, among which 105 were differentially expressed following restraint stress (fold change ≥ 2.0, *p* < 0.05), including 30 up-regulated and 51 down-regulated proteins (Figure 2a,b, Appendix A). GO enrichment analysis showed that these differentially expressed proteins were significantly involved in immune-related pathways (e.g., B cell–mediated immunity, adaptive immune response), chromatin organization (e.g., nucleosome assembly, DNA packaging complex), and neuroendocrine signaling (e.g., insulin receptor activity) (Figure 2c,d). Kyoto Encyclopedia of Genes and Genomes (KEGG) pathway analysis further revealed significant alterations in energy metabolism (e.g., citrate cycle, TCA cycle), inflammation (e.g., NF-κB signaling), and immune regulation (e.g., antigen presentation). Notably, enrichment of β-alanine metabolism (*p* < 0.01) and the p53 signaling pathway (*p* < 0.05) suggests that neurotransmitter imbalance and cellular stress responses may contribute to the observed depression-like behaviors (Figure 2e).

Up-regulated proteins were enriched in pathways related to synaptic function, transcriptional repression, and DNA repair, including regulation of postsynaptic membrane potential (e.g., GABAA receptor subunit α6 [GABRA6], potassium channel KCNK1), RNA polymerase II-mediated repression (Histone H3C6), and DNA damage response (Zinc finger protein ZMAT3) (Appendix A). Additionally, down-regulated proteins were associated with immunoglobulin-mediated immunity (e.g., IGHV/IGKV), mitochondrial uncoupling and redox balance (e.g., uncoupling protein 2 [UCP2]), and membrane hyperpolarization involved in neuronal excitability (e.g., potassium channel KCNQ3, Glucose transporter SLC2A4) (Appendix A).

To investigate the functional organization of stress-responsive proteins, we constructed a protein–protein interaction (PPI) network using differentially expressed candidates (Figure 2f). The resulting network highlighted several interconnected modules associated with epigenetic regulation (e.g., chromobox protein homolog 8 [CBX8], H3C6, protein inhibitor of activated STAT 3 [PIAS3], histone deacetylase 1 [HDAC1]), mitochondrial and ribosomal components (e.g., mitochondrial ribosomal protein L51 [MRPL51], ribosomal protein L7-like 1 [RPL7L1]), and synaptic signaling (e.g., GABRA6, rho guanine nucleotide exchange factor 33 [ARHGEF33]). Notably, chromatin-associated factors such as CBX8 and AT-rich interactive domain-containing protein 4 (AARID4A) formed a dense interaction cluster, suggesting a coordinated regulatory mechanism in chromatin remodeling. Additionally, immune-related proteins including S100 calcium-binding protein A8 (S100A8) and complement component 4-binding protein (C4BP) were embedded within immune response sub-modules. These findings reveal a structured proteomic response in the mPFC, linking transcriptional repression, metabolic regulation, and immune signaling under acute stress conditions.

### 3.3. Integrated Omics Reveal Coordinated Changes in Epigenetic Regulation and Immune Response

Integrated transcriptomic and proteomic analyses identified 31 genes consistently up-regulated and 9 down-regulated at both mRNA and protein levels (fold change ≥ 1.2, *p* < 0.05) (Figure 3a). Notably, these genes span several functionally distinct categories, suggesting long-lasting molecular adaptations that involve coordinated changes across epigenetic, metabolic, immune, and neurodevelopmental pathways (Figure 3b–d).

Several chromatin-associated and transcriptional regulatory factors were upregulated at both mRNA and protein levels, including H3c6 (a replication-dependent histone H3 variant), Smarcd2 (a SWI/SNF chromatin remodeler), Rbm39 (an RNA-binding splicing coactivator), Med8 (Mediator complex subunit), and Zrsr2 (a spliceosomal regulator). These alterations point to sustained transcriptional reorganization and support the hypothesis that acute stress may lead to long-lasting epigenetic adaptations and gene expression control in the mPFC, potentially contributing to the maintenance of depressive-like behaviors.

Upregulation of Il33 (an alarmin cytokine), Cuedc2 (a regulator of NF-κB and cytokine signaling), and Rell1 (a TNF-receptor-like molecule) suggests activation of neuroimmune pathways in the mPFC. These factors may contribute to sustained low-grade inflammation or glial activation, which are increasingly recognized as contributors to stress-induced behavioral alterations and synaptic dysfunction.

Collectively, these concordant transcriptome-proteome changes reflect a broad, multi-system remodeling of the mPFC that persists weeks after acute stress exposure. They highlight the integration of epigenetic plasticity, mitochondrial bioenergetics, inflammatory signaling, and neurostructural reprogramming as key features of stress-induced long-term vulnerability or adaptation.

### 3.4. Proteomic Profiling Reveals Clemastine-Mediated Restoration of Inflammatory and Metabolic Dysregulation in the mPFC

Clemastine, traditionally known for its anti-allergic properties via H1 receptor antagonism [19,20], has emerged as a multi-functional small molecule with central nervous system effects. Beyond its classical peripheral actions, clemastine promotes oligodendrocyte differentiation and remyelination by inhibiting M1/M3 muscarinic receptors [21,22], and exerts potent anti-inflammatory effects by suppressing microglial activation [23,24]. These pharmacological properties have been linked to neuroprotective benefits in models of multiple sclerosis [25,26], Alzheimer’s disease [27,28], and depression [17].

To evaluate clemastine’s therapeutic effects, we treated restraint-stressed mice with clemastine and conducted proteomic analysis on the mPFC. After administering 10 mg/kg clemastine via drinking water for 33 days, reduced sucrose preference and increased immobility in FST were rescued in treatment group (Figure 4a,b). Temporal clustering analysis identified two notable clusters: cluster 7 contained proteins that were down-regulated by restraint stress and restored by clemastine treatment, while cluster 9 included proteins up-regulated by stress but normalized following clemastine administration (Figure 4c).

GO enrichment analysis of cluster 7 revealed significant associations with innate immune responses (e.g., Toll-Like Receptor 4, TLR4 signaling pathway), mitochondrial matrix functions and zinc ion binding (Figure 5a,b and Appendix A). Clemastine restored the expression of genes that were suppressed under stress (cluster 7), including mitochondrial enzymes (Ehhadh, Cox6a1), synaptic regulators (Tac1, Farp2), and immune mediators (Cd14, Nfkb2). It suggests that clemastine may exert neuroprotective effects by simultaneously modulating inflammation and oxidative stress.

GO enrichment analysis of cluster 9 showed that significant associations with potassium ion transmembrane transport, regulation of postsynaptic membrane potential, and protein N-linked glycosylation, suggesting alterations in ion homeostasis and synaptic signaling under stress conditions (Figure 5c,d and Appendix A). It suggests that clemastine may stabilize neuronal activity by attenuating stress-induced alterations in ion channel expression or function, particularly those associated with inward rectifier potassium channels, which are critical for maintaining resting membrane potential and limiting excitotoxicity.

Specifically, clemastine downregulated genes that were pathologically overexpressed following stress, including histone H3 variants (H3C6) and chromatin regulators (Smarcd2, Arid4a). Additionally, the enrichment of chromatin-related components such as the nucleoplasm, chromatin, and SWI/SNF complex implies a role for clemastine in influencing transcriptional regulation. SWI/SNF complexes are ATP-dependent chromatin remodelers involved in regulating gene expression in response to environmental stimuli, including stress. Dysregulation of chromatin remodeling has been implicated in the pathophysiology of depression and stress-induced behavioral phenotypes. Thus, clemastine may exert protective effects in part by restoring epigenetic homeostasis in prefrontal cortical neurons. This aligns with clemastine’s reported ability to modulate chromatin accessibility and transcriptional output in glial and neuronal cells [29,30].

## 4. Discussion

In our study, transcriptomic analysis revealed that acute stress significantly disrupted dopaminergic synaptic signaling and oxidative stress responses in the mPFC. Proteomic profiling revealed broader changes not captured at the transcript level, including alterations in chromatin organization and immune signaling. Despite low global concordance, transcriptomic and proteomic datasets revealed overlapping pathways involving immune responses and metabolic regulation. Importantly, clemastine, a clinically available H1 receptor antagonist, effectively rescues stress-induced molecular alterations in the mPFC. Proteins involved in NF-κB signaling, chromatin remodeling, and energy metabolism were normalized, indicating its potential role in mitigating neuroinflammation, epigenetic dysregulation, and metabolic imbalance associated with depression.

Transcriptomic and proteomic analyses both identified reduced NF-κB signaling activity, a central pathway in neuroinflammation. This suppression may reflect an initial adaptive response to acute stress but could also impair neuroimmune homeostasis and contribute to synaptic and metabolic dysfunction over time. Stress induces the hypothalamic–pituitary–adrenal (HPA) axis to release the neuroendocrine hormones, which have been shown to be involved in activating immune cells and promoting the release of inflammatory mediators [7,31]. The regulation between immune cells and neurohormones is also bidirectional, as inflammatory cytokines also activate the HPA axis [32].

Clemastine’s reversal of stress-induced immune suppression may play a central role in its neuroprotective effects. GO enrichment of clemastine-responsive genes revealed significant associations with innate immune signaling, particularly TLR4-related pathways, as well as NF-κB regulation—an axis critical for maintaining neuroimmune homeostasis. By restoring IκBα and Nfkb2 expression, clemastine may help reestablish proper NF-κB signaling dynamics, counteracting the stress-induced immunosuppression that can compromise neuronal resilience. Additionally, normalization of zinc finger proteins and mitochondrial enzymes such as Cox6a1 and Ehhadh suggests that clemastine mitigates oxidative stress and preserves mitochondrial function—both of which are closely linked to inflammatory status and synaptic health. These findings are consistent with previous reports that clemastine mediates its therapeutic effect through inhibition of microglia-induced inflammatory response and apoptosis, thereby enhancing restoration of neuronal function [33]. Thus, clemastine appears to exert antidepressant-like effects, at least in part, by restoring immune competence and preventing sustained neuroinflammatory damage in the stressed brain.

In addition to the medial prefrontal cortex, the hippocampus and other cognition-related regions are critically involved in depression. Numerous studies have shown that chronic stress impairs hippocampal neurogenesis and synaptic plasticity, leading to memory and learning deficits [34,35]. Such cognitive symptoms are increasingly recognized as a core component of major depressive disorder, beyond mood disturbances. Moreover, depression and chronic stress have been identified as risk factors for dementing disorders, including Alzheimer’s disease, suggesting a mechanistic link through neuroinflammation [36], mitochondrial dysfunction [37], and impaired synaptic resilience [4,38]. Although our present work focused on the mPFC, future studies should investigate whether clemastine also confers protective effects on hippocampal-dependent cognitive functions and neurodegenerative risk.

Although our integrated transcriptomic and proteomic analyses provide novel insights into the molecular mechanisms of stress-induced pathology and clemastine’s therapeutic effects, several limitations should be considered. First, the single timepoint design precludes assessment of dynamic molecular changes during stress progression and recovery. Another limitation of our study is the relatively small number of biological replicates (*n* = 3 per group) used for RNA-seq and proteomic analyses. While this sample size is commonly adopted in exploratory multi-omics studies, it is underpowered for comprehensive discovery and may limit the ability to detect subtle molecular changes. Larger cohorts and independent validation experiments (e.g., qPCR or targeted proteomics) will be essential to confirm and extend these findings in future studies. Finally, species differences between mice and humans should be carefully considered. For example, clemastine’s metabolic rate, pharmacokinetics, and neuroimmune modulation may differ substantially in humans, which could influence its therapeutic efficacy. Therefore, while our findings provide important mechanistic insights, future translational and clinical studies will be necessary to validate the antidepressant potential of clemastine in patients. Future studies incorporating longitudinal designs, cell-type-specific approaches, and human translational models will be important to address these limitations.

## Figures and Tables

**Figure 1 biomedicines-13-02183-f001:**
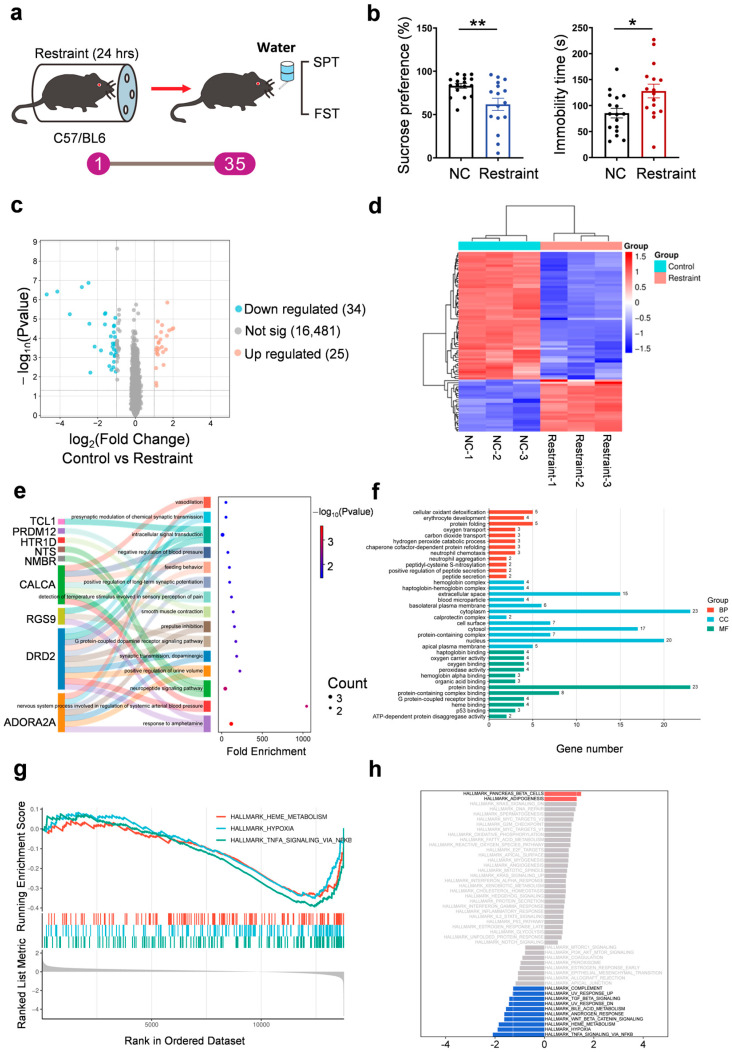
Transcriptomic profiling. (**a**) Diagram of the 24-hour restraint stress mouse model. (**b**) Quantification of the sucrose preference in SPT (** *p* = 0.007) and immobility time in 4 min in FST (* *p* = 0.0121, *n* = 17 for NC group and *n* = 16 for restraint group, each dot represents one mouse). (**c**) Volcano plot for differentially expressed genes in restraint mice samples compared with controls in RNA-seq. Blue and red dots indicate statistical down- and up-regulated genes, respectively. (**d**) Heat-map analysis on expressions of genes in the mPFC (*n* = 3 for each group). (**e**,**f**) GO—‘Biological Processes’/‘Cellular Component’/‘Molecular Function’ analysis on up-regulated genes in restraint mice compared with control mice in RNA-seq. (**g**,**h**) GSEA analysis of differentially expressed genes in restraint mice samples compared with controls in RNA-seq. Red indicates gene sets positively enriched in the stress group (NES > 0), while blue indicates gene sets negatively enriched in stress (NES < 0). NES, normalized enrichment score.

**Figure 2 biomedicines-13-02183-f002:**
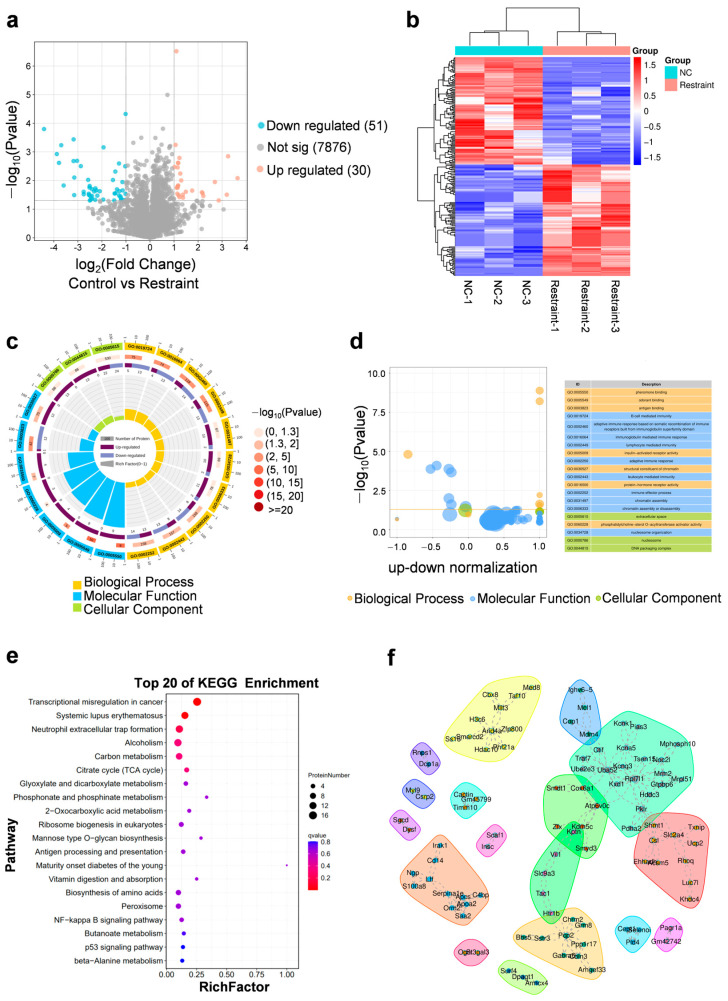
Proteomic profiling. (**a**) for differentially expressed proteins in restraint mice samples compared with controls in proteomics. Blue and red dots indicate statistical down- and up-regulated proteins, respectively. (**b**) Heat-map analysis on expressions of proteins in the mPFC (*n* = 3 for each group). (**c**,**d**) GO—‘Biological Processes’/‘Cellular Component’/‘Molecular Function’ analysis on differentially expressed proteins in restraint mice compared with control mice in proteomics. (**e**,**f**) KEGG analysis (**e**) and PPI analysis (**f**) of differentially expressed proteins in restraint mice samples compared with controls in proteomics.

**Figure 3 biomedicines-13-02183-f003:**
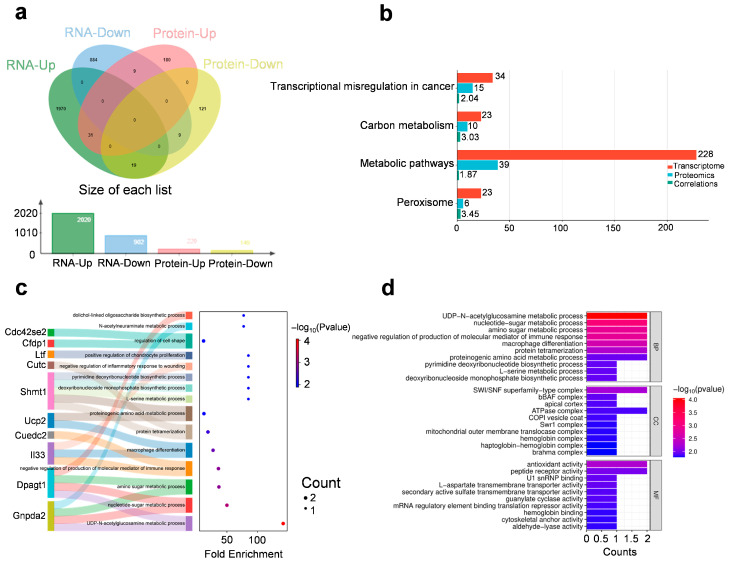
Integrated transcriptomic and proteomic analysis. (**a**) The distribution of differentially expressed genes in transcriptomics and proteomics. (**b**) Shared KEGG pathways enriched in both transcriptomic and proteomic datasets. The bar plot shows the number of genes enriched in selected KEGG pathways from transcriptomic (red) and proteomic (blue) analyses. Cyan bars represent the average gene-level correlation coefficients between mRNA and protein expression within each pathway. (**c**,**d**) GO—‘Biological Processes’/‘Cellular Component’/‘Molecular Function’ analysis on overlapping up-regulated genes in transcriptomics and proteomics.

**Figure 4 biomedicines-13-02183-f004:**
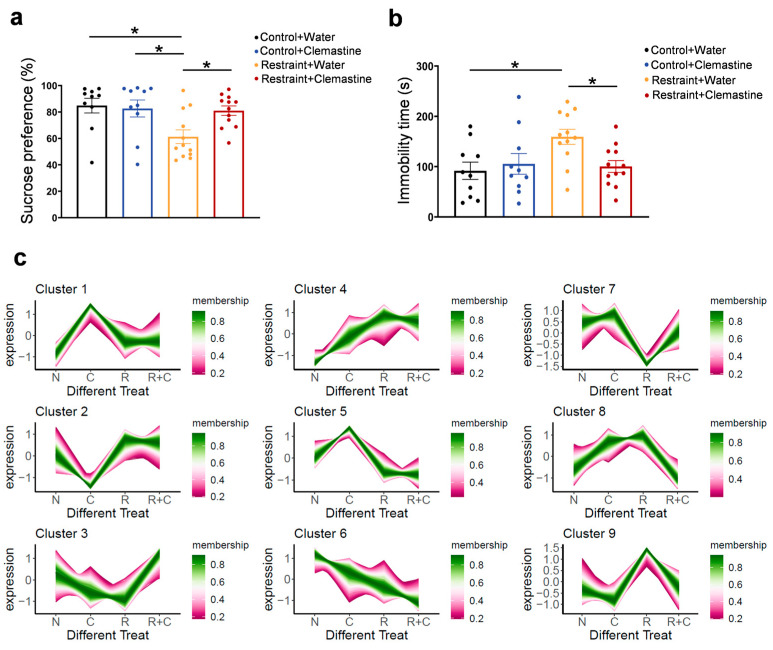
Clemastine treatment alleviates depressive-like behavior. (**a**,**b**) Behavioral assessments show that clemastine treatment restores sucrose preference ((**a**), Control + Water versus Restraint + Water, * *p* = 0.0155; Restraint + Water versus Restraint + Clemastine, * *p* = 0.0439; *n* = 10 for Control group and *n* = 12 for restraint group) and reduces immobility time ((**b**), Control + Water versus Restraint + Water, * *p* = 0.0297; Restraint + Water versus Restraint + Clemastine, * *p* = 0.0452; *n* = 10 for Control group and *n* = 12 for restraint group) in restraint-stressed mice, indicating antidepressant-like effects. Each dot represents one mouse. (**c**) Temporal clustering analysis of proteomics datasets of mPFC tissues from mice (N means control group, C means control + clemastine group, R means restraint group, R + C means restraint+clemastine group, *n* = 3 for each group).

**Figure 5 biomedicines-13-02183-f005:**
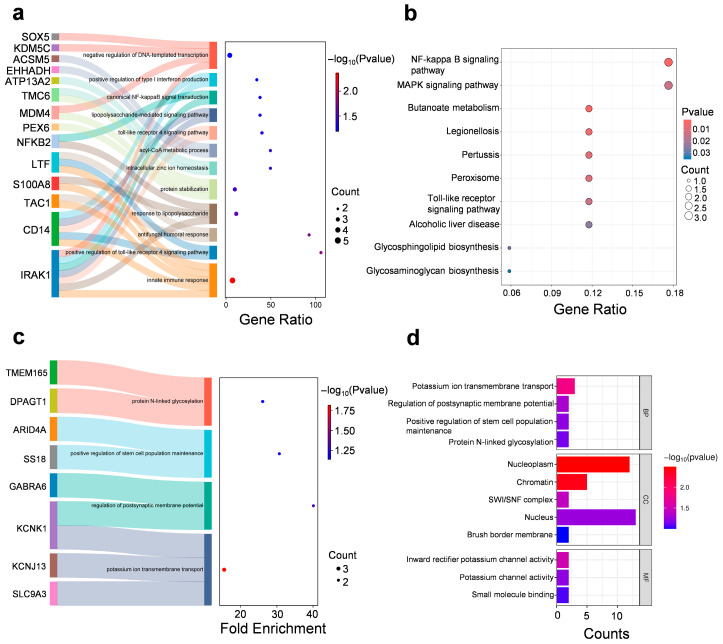
Clemastine treatment alleviates depressive-like behavior and reverses stress-induced proteomic alterations in the mPFC. (**a**) GO—‘Biological Processes’ analysis on cluster 7 included proteins. (**b**) KEGG analysis on cluster 7 included proteins. (**c**,**d**) GO—‘Biological Processes’/‘Cellular Component’/‘Molecular Function’ analysis on cluster 9 included proteins.

## Data Availability

All data reported in this paper will be shared by the lead contact upon request.

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
