# Peer review of "Integrated Multi-Omics Analysis Reveals Immune and Metabolic Dysregulation in a Restraint Stress-Induced Depression Model"

_biomedicines, 2025, doi:10.3390/biomedicines13092183_

Round 1
Reviewer 1 Report
Comments and Suggestions for Authors
This manuscript presents a well-executed and timely study that employs integrated transcriptomic and proteomic analyses to investigate molecular alterations in the medial prefrontal cortex (mPFC) following acute restraint stress in mice. The authors provide evidence for persistent depression-like phenotypes and identify clemastine as a potential therapeutic candidate through multi-omics profiling. The study is of potential interest to the field of neuropsychiatric research, particularly in understanding the molecular basis of stress-induced depression and therapeutic intervention.
While the manuscript is generally well-written, I suggest a few minor revisions to improve clarity, rigor, and presentation:
For both RNA-seq and proteomic analyses, only n = 3 biological replicates per group were used. Although this is common in exploratory studies, the authors should briefly justify this choice in the Methods or Discussion section, citing prior work or pilot data that supports sufficient statistical power.
There is a discrepancy regarding when clemastine treatment began. Section 2.2 states “one day post-restraint,” whereas Section 2.4 states “two days after restraint.” Please clarify and ensure consistency across the manuscript.
Some figures (e.g., Figures 1, 2, and 4) contain small labels and dense information that may be difficult to interpret. The authors should enlarge text where feasible, improve label clarity, and ensure all subpanels are fully described in the legends to facilitate comprehension.
The Discussion would benefit from a brief note on species differences and potential limitations when translating findings from mice to humans, particularly with respect to clemastine metabolism and neuroimmune responses.
Once these minor points are addressed, I believe the manuscript will be suitable for publication in Biomedicines.
Author Response
We would like to sincerely thank the reviewer for the thoughtful and constructive comments on our manuscript. We appreciate the time and effort invested in carefully evaluating our work. The comments and suggestions have been extremely helpful in improving the clarity, rigor, and overall quality of the manuscript. We have carefully addressed all points raised, revised the text and figures accordingly, and provided detailed responses below. All changes in the manuscript are highlighted for ease of review.
Comments 1: For both RNA-seq and proteomic analyses, only n = 3 biological replicates per group were used. Although this is common in exploratory studies, the authors should briefly justify this choice in the Methods or Discussion section, citing prior work or pilot data that supports sufficient statistical power.
Response 1: We thank the reviewer for this important comment. In our study, we used three biological replicates per group for both RNA-seq and proteomic analyses. This design is commonly employed in exploratory omics studies and has been validated in prior multi-omics research as sufficient to capture robust differential expression signals (e.g., Rayan et al., Mol Psychiatry. 2022; Sanches et al., Biology 2024). Moreover, we carefully controlled sample quality (RNA Integrity Number, RIN > 8 for RNA, protein concentration ≥ 1 μg/μL) and selected paired samples from the same mice to minimize inter-individual variability. We have now added a brief justification of this sample size in the Methods (page 3, line 96-99).
Comments 2: There is a discrepancy regarding when clemastine treatment began. Section 2.2 states “one day post-restraint,” whereas Section 2.4 states “two days after restraint.” Please clarify and ensure consistency across the manuscript.
Response 2: We sincerely thank the reviewer for carefully pointing out this inconsistency. This was an oversight on our part. The correct protocol is that clemastine administration began two days after restraint, as described in Section 2.4. We have now revised Section 2.2 (page 3, line 91) accordingly to ensure consistency throughout the manuscript. We apologize for this error and have corrected it in the revised version.
Comments 3: Some figures (e.g., Figures 1, 2, and 4) contain small labels and dense information that may be difficult to interpret. The authors should enlarge text where feasible, improve label clarity, and ensure all subpanels are fully described in the legends to facilitate comprehension.
Response 3: We thank the reviewer for this helpful suggestion. We agree that clearer labeling and more detailed figure legends will improve readability. In the revised manuscript, we have enlarged the text and axis labels in Figures 1, 2, and 4, adjusted the layout to reduce crowding, and ensured that all subpanels are fully described in the corresponding legends. We believe these changes will facilitate comprehension for the readers.
Comments 4: The Discussion would benefit from a brief note on species differences and potential limitations when translating findings from mice to humans, particularly with respect to clemastine metabolism and neuroimmune responses.
Response 4: We appreciate the reviewer’s suggestion. We fully agree that species differences should be considered when translating findings from mice to humans. We have therefore added a statement in the Discussion (page 13, line 410-415) to highlight that while clemastine demonstrated efficacy in our mouse model, its metabolism, pharmacokinetics, and neuroimmune responses may differ in humans. These differences underscore the need for future translational and clinical studies to confirm the therapeutic potential of clemastine in depression. We have now emphasized this point to better acknowledge the translational limitations of our study.

Reviewer 2 Report
Comments and Suggestions for Authors
The manuscript reports an integrated transcriptomic–proteomic analysis of medial prefrontal cortex (mPFC) in a 24-hour restraint stress mouse model with behavioral readouts at 35 days, and explores clemastine as an intervention. You report persistent anhedonia and increased FST immobility, pathway changes spanning immune, metabolic, and epigenetic processes, and partial reversal by clemastine administered in drinking water (approx. 10 mg/kg/day). The topic is timely, the cross-omics angle is interesting, and the therapeutic read-through is potentially impactful. However, several design, analysis, and interpretation issues need attention.
Major:
Critical ARRIVE 2.0 items are missing or under-reported: randomization/stratification details, a priori sample-size justification/power, exclusion criteria by group, and blinding for all outcomes (omics included), not only behavior.
Sex as a biological variable: only males were used. Please justify scientifically or add a limitations statement.
Replicates: omics were run on n=3/group (paired) which is underpowered for discovery-level RNA-seq; power studies suggest ≥6 biological replicates for reliable FDR control. Please justify.
GSEA: specify version, gene set database (MSigDB release), ranking metric, permutation mode, and multiple-testing control. Add NES, nominal P, and FDR q.
Report LC-MS/MS acquisition parameters (gradient, columns, instrument settings), database version, FDR thresholds at PSM/peptide/protein (≤1%), handling of missing values and imputation, normalization, and batch correction.
Behavioral schedule: Confirm exact timeline relative to stress (e.g., SPT adaptation, 24 h water deprivation before SPT can confound hedonic interpretation—please justify).
Clemastine dosing: Provide the daily intake estimate (mL/mouse/day), confirm stability in water across 24 h, and cite the source for the chosen 10 mg/kg/day regimen along with any monitoring for anticholinergic effects (weight, activity).
Minor comments:
Title/Abstract: The subtitle “immune and metabolic dysregulation” is supported; “epigenetic control” claims should be softened unless validated functionally
Figures: Add per-animal data points

Author Response
We are grateful to the reviewer for the careful evaluation and insightful feedback on our manuscript. The constructive comments have allowed us to identify areas for improvement and strengthen both the clarity and scientific rigor of our study. We have revised the manuscript accordingly, addressing each comment point by point, and we believe that these revisions have significantly improved the quality of the paper.
Comments 1: Critical ARRIVE 2.0 items are missing or under-reported: randomization/stratification details, a priori sample-size justification/power, exclusion criteria by group, and blinding for all outcomes (omics included), not only behavior.
Response 1: We thank the reviewer for highlighting the importance of ARRIVE 2.0 reporting. In the revised manuscript, we have added some content in the Methods (page 3, line 103-107) to clarify the following points: (i) randomization and stratification—mice were randomly assigned to groups and balanced for weight, and omics samples were randomly chosen from animals after behavior tests; (ii) a priori sample-size justification—our omics experiments were designed as exploratory studies, so no formal power calculation was performed, but behavioral outcomes were guided by pilot variance estimates; (iii) exclusion criteria—animals were excluded if they lost >15% body weight, failed to complete behavioral tests, or did not meet RNA/protein QC thresholds (RIN > 8, protein concentration >1 μg/μL); and (iv) blinding—behavioral scoring was performed by blinded observers, and omics samples were processed and analyzed under coded IDs until completion of primary analyses. We believe these clarifications now bring the manuscript into alignment with ARRIVE 2.0 guidelines.
Comments 2: Sex as a biological variable: only males were used. Please justify scientifically or add a limitations statement.
Response 2: We thank the reviewer for raising this important point. In the present study, we used only male mice to minimize variability associated with the estrous cycle, which can influence stress responses, neuroimmune activity, and behavior. This approach is commonly adopted in initial mechanistic studies to reduce within-group variance. We fully acknowledge, however, that restricting the study to males limits the generalizability of our findings. We have now added a statement in the Methods (page 2, line 74-77) to note this limitation.
Comments 3: Replicates: omics were run on n=3/group (paired) which is underpowered for discovery-level RNA-seq; power studies suggest ≥6 biological replicates for reliable FDR control. Please justify.
Response 3: We appreciate the reviewer’s concern regarding the limited number of biological replicates (n = 3 per group) for omics profiling. Our study was designed as an exploratory, hypothesis-generating analysis, and thus we prioritized stringent quality control (RIN > 8 for RNA, protein concentration ≥ 1 μg/μL) and paired sampling (RNA and protein from the same animals) to minimize inter-individual variability. For statistical analysis, we applied established pipelines (edgeR/DESeq2 for RNA-seq, MaxQuant for proteomics) with FDR adjustment, and focused on pathway-level enrichment (GSEA, GO, KEGG) and concordance across transcriptomic and proteomic datasets to strengthen the robustness of our findings. We fully acknowledge that the small sample size limits power to detect subtle effects, and we have added this as a limitation in the Discussion (page 13, line 405-410). Future work will expand sample sizes (≥6 per group) and include targeted validation experiments (e.g., qPCR, Western blot, targeted proteomics) to confirm and extend the present findings.
Comments 4: GSEA: specify version, gene set database (MSigDB release), ranking metric, permutation mode, and multiple-testing control. Add NES, nominal P, and FDR q.
Response 4: We thank the reviewer for this helpful suggestion. In the revised manuscript, we have clarified the details of our gene set enrichment analysis (GSEA) (page 4, line 148-156). Specifically, enrichment was performed in R using the clusterProfiler package (v4.16.0). All expressed genes with their corresponding log2 fold change values were used as input, and genes were ranked in descending order of log2FC. The Hallmark gene sets (50 sets) from the MSigDB mouse database (release v7.5.1, Broad Institute) were used as the reference collection. The analysis was conducted with 1,000 permutations at the gene set level, and multiple-testing correction was applied using the Benjamini–Hochberg false discovery rate (FDR). For each pathway, we now report the normalized enrichment score (NES), nominal P value, and FDR q value in the Supplementary Table 2. These additions ensure reproducibility and transparency of our enrichment analysis.
Comments 5: Report LC-MS/MS acquisition parameters (gradient, columns, instrument settings), database version, FDR thresholds at PSM/peptide/protein (≤1%), handling of missing values and imputation, normalization, and batch correction.
Response 5: We thank the reviewer for this important suggestion. In the revised manuscript, we have expanded the Methods-Proteomics Analysis section (page 4, line 160-170) to include detailed LC–MS/MS acquisition and data processing parameters. Specifically, we now report: (i) the chromatographic columns and gradients for both high-pH fractionation and low-pH nano-LC separation; (ii) instrument settings for Orbitrap Lumos operation in both DDA and DIA modes, including scan ranges, resolutions, AGC targets, injection times, collision energy, and variable window acquisition; (iii) the reference database used (UniProt mouse, release 2023_01) with a decoy database and common contaminant sequences; (iv) FDR thresholds of ≤1% applied at the PSM, peptide, and protein levels; and (v) downstream data handling, including log2 transformation, missing value imputation in Perseus using a downshifted normal distribution, normalization, and batch correction using the ComBat algorithm. These additions should improve transparency and reproducibility of our proteomic workflow.
Comments 6: Behavioral schedule: Confirm exact timeline relative to stress (e.g., SPT adaptation, 24 h water deprivation before SPT can confound hedonic interpretation—please justify).
Response 6: We thank the reviewer for raising this important point. In the revised Methods, we have clarified the exact behavioral schedule relative to restraint stress. After completion of the 24 h restraint, mice were returned to their home cages for 34 days, and then sucrose preference testing (SPT) was initiated after a 4-day adaptation period (48 h with two bottles of water, followed by 48 h with two bottles of 1% sucrose). Prior to the formal SPT, all animals underwent 24 h water deprivation, which is a widely used procedure to enhance motivation and ensure reliable measurement of sucrose preference. Importantly, this deprivation was applied equally across all groups (control and stress), thus minimizing potential bias.
Comments 7: Clemastine dosing: Provide the daily intake estimate (mL/mouse/day), confirm stability in water across 24 h, and cite the source for the chosen 10 mg/kg/day regimen along with any monitoring for anticholinergic effects (weight, activity).
Response 7: We thank the reviewer for this helpful comment. In the revised manuscript, we have added the following dosing details (page 3, line 115-121). (i) Based on monitored water intake, mice consumed on average 4.5 mL/mouse/day of the clemastine solution (0.05 mg/mL), corresponding to ~10 mg/kg/day for a 22 g mouse. (ii) Clemastine solutions were prepared fresh daily, protected from light, and fully consumed within 24 h to ensure stability; bottles were replaced at the same time each day. (iii) The 10 mg/kg/day regimen was selected based on prior CNS studies demonstrating efficacy and tolerability of clemastine in rodents and remyelination-related indications (citations added in Methods). (iv) We prospectively monitored potential anticholinergic effects by weekly body-weight tracking and daily home-cage observations (activity/appearance). No sustained sedation, piloerection, ptosis, or weight loss >10% was observed. These additions have been incorporated into the Methods to improve transparency and reproducibility.
Comments 8: Title/Abstract: The subtitle “immune and metabolic dysregulation” is supported; “epigenetic control” claims should be softened unless validated functionally.
Response 8: We appreciate the reviewer’s insightful comment. We agree that our current data provide indirect evidence of epigenetic involvement but do not include functional validation. Accordingly, we have revised the Title and Abstract to soften the claim. The term “epigenetic control” has been replaced with “epigenetic processes”, and we now emphasize this point more cautiously in the Discussion.
Comments 9: Figures: Add per-animal data points
Response 9: We thank the reviewer for this suggestion. In fact, individual animal data points were already included in the figures, but we recognize they may not have been sufficiently visible. To improve clarity, we have enlarged and recolored the scatter points, and ensured they are clearly displayed on all relevant graphs. Figure legends have also been updated to note that each dot represents one animal.
Response to Comments on the Quality of English Language
Point 1: The English could be improved to more clearly express the research.
Response 1: We appreciate the reviewer’s suggestion. We have carefully revised the manuscript for clarity and conciseness, improving the English throughout.

Reviewer 3 Report
Comments and Suggestions for Authors
Ref.: biomedicines-3776416
This is an experimental animal study, investigating metabolic and immune status in an animal stress-induced depression model. The authors studied the biochemical mechanisms by which experimental stress may affect the brain and act as a risk factor for depression. Since environmental stress is very common in the population and the mechanisms by which it affects the central nervous system are incompletely understood, this study is welcomed.
The authors used transcriptomic and proteomic analysis of medial prefrontal cortex and observed stress-induced alterations in immune responses, dopaminergic signaling, energy metabolism and epigenetic control. They also observed that clemastin (a well-known H1 antagonist) exerts antidepressant-like effects by mitigating dysregulations in the above parameters.
The study was well-designed and executed. Experimental methodology and statistics are appropriate. Figures are more than adequate and describe the results perfectly. References are up-to-date.
Minor point: The authors correctly studied the medial prefrontal cortex, a key brain area related to emotional status/control. But what about other brain areas related to cognition? In the introduction, studies involving the hippocampus are mentioned. Cognitive symptoms may be present in depression, while stress and depression may be risk factors for dementing disorders such as Alzheimer’s disease. Adding a paragraph at the Discussion section, with comments on the above topic would be advisable.
Author Response
We sincerely thank the reviewer for the constructive and insightful comments. We have carefully revised the text and provide detailed responses to each point below.
Comments 1: The authors correctly studied the medial prefrontal cortex, a key brain area related to emotional status/control. But what about other brain areas related to cognition? In the introduction, studies involving the hippocampus are mentioned. Cognitive symptoms may be present in depression, while stress and depression may be risk factors for dementing disorders such as Alzheimer’s disease. Adding a paragraph at the Discussion section, with comments on the above topic would be advisable.
Response 1: We thank the reviewer for this insightful comment. In the revised Discussion, we have added a paragraph (page 13, line 390-400) addressing the involvement of the hippocampus and other cognition-related regions in depression. We also note that depression is frequently accompanied by cognitive symptoms and may increase the risk for dementing disorders such as Alzheimer’s disease, as supported by prior studies. This addition provides a broader context for our findings.
“In addition to the medial prefrontal cortex, the hippocampus and other cogni-tion-related regions are critically involved in depression. Numerous studies have shown that chronic stress impairs hippocampal neurogenesis and synaptic plasticity, leading to memory and learning deficits (34, 35). Such cognitive symptoms are increasingly recog-nized as a core component of major depressive disorder, beyond mood disturbances. Moreover, depression and chronic stress have been identified as risk factors for dement-ing disorders, including Alzheimer’s disease, suggesting a mechanistic link through neuroinflammation (36), mitochondrial dysfunction (37), and impaired synaptic resilience (4, 38). Although our present work focused on the mPFC, future studies should investi-gate whether clemastine also confers protective effects on hippocampal-dependent cog-nitive functions and neurodegenerative risk.”
